# Crop Productivity Boosters: Native Mycorrhizal Fungi from an Old-Growth Grassland Benefits Tomato (*Solanum lycopersicum*) and Pepper (*Capsicum annuum*) Varieties in Organically Farmed Soils

**DOI:** 10.3390/microorganisms11082012

**Published:** 2023-08-04

**Authors:** Liz Koziol, James D. Bever

**Affiliations:** Kansas Biological Station and Ecology and Evolutionary Biology, University of Kansas, Lawrence, KS 66047, USA

**Keywords:** annual crops, arbuscular mycorrhizal fungi, biofertilizers, cultivars, genotypic and phenotypic variation, inoculation, organic farming, vegetables, symbiosis

## Abstract

This paper investigates the response of five tomato and five pepper varieties to native arbuscular mycorrhizal (AM) fungal inoculation in an organic farming system. The field experiment was conducted across a growing season at a working organic farm in Lawrence, KS, USA. The researchers hypothesized that native AM fungi inoculation would improve crop biomass production for both crop species, but that the magnitude of response would depend on crop cultivar. The results showed that both crops were significantly positively affected by inoculation. AM fungal inoculation consistently improved total pepper biomass throughout the experiment (range of +2% to +8% depending on the harvest date), with a +3.7% improvement at the final harvest for inoculated plants. An interaction between pepper variety and inoculation treatment was sometimes observed, indicating that some pepper varieties were more responsive to AM fungi than others. Beginning at the first harvest, tomatoes showed a consistent positive response to AM fungal inoculation among varieties. Across the experiment, AM fungi-inoculated tomatoes had +10% greater fruit biomass, which was driven by a +20% increase in fruit number. The study highlights the potential benefits of using native AM fungi as a soil amendment in organic farmed soils to improve pepper and tomato productivity.

## 1. Introduction

Maintaining long-term field productivity in agricultural soils requires continuous effort and inputs for farms and nurseries of all scales. Maintaining productive soils in organic agriculture is even more challenging, as fewer options are available for farmers to improve the abiotic and biotic soil conditions of their fields. A principal challenge is building and maintaining nutrient-rich soil without inorganic inputs. Organic systems often have lower nutrient additions and, ultimately, lower available phosphorus and other nutrients [1,2] relative to conventionally managed lands that rely on inorganic inputs. Amending soils with organic matter such as compost and manure can improve organic crop productivity [2], but these practices require considerable effort and expense. To mitigate these costs, some organic growers have begun to embrace a widely accepted understanding of natural systems; soil microbes are the drivers of soil health and nutrient cycling [3,4]. Thus, more farmers are interested in managing their soil microbes.

Key members of the soil community that are central to improving soil health are arbuscular mycorrhizal (AM) fungi. AM fungi are microscopic microbes that spend their entire lives in soil. Around 80% of plants associate with mycorrhizae in a symbiotic relationship where fungi collect soil nutrients for plants, and in exchange, plants release sugars from their roots to feed the fungi. Most crop species belong to plant families that associate with AM fungi [5], and many annual [6,7,8,9] and perennial [10,11] crops have been shown to benefit from mycorrhizal amendments in the field, including grains, fruits, vegetables, and oil seed crops. Although AM fungi are commonly present in soils, AM fungal density, diversity, and composition in agricultural environments may be limited by site history. Many agricultural systems include land manipulations known to disrupt fungal communities, such as tilling [12,13], the use of soluble fertilizers and biocides [14], and the planting of monocultures [15].

Because agricultural disturbance can result in ineffective AM fungal communities, one approach to the management of beneficial soil microorganisms is to add microbial amendments back into organic soils with a history of soil disturbance. Many microbial inoculation studies have tested commercial inoculants, but several studies have compared native and commercial inoculants of non-native origin. Native AM fungi inoculations have been found to be more effective or as effective as commercial inoculants for corn [16], tropical trees [17], and other crops [18,19,20]. Because mycorrhizal fungi can be locally adapted to their home soil nutrient and precipitation levels [21,22], some farmers have turned to native AM fungal amendments in their fields. Past research has found that native AM fungal inoculations can improve pepper transplant success relative to non-native inocula [19]. Others have found that native AM fungi can improve tomato resistance to root-knot nematode attack [18]. Overall, native AM inoculation studies indicate that adding AM fungi to fields can improve the growth of many crops, including *Solanaceae*, *Leguminosae*, and *Cucurbitaceae* [23]. However, one limitation of past studies is that they often do not occur in organically managed systems; those that do have often occurred in European or African soils [16,17,18,19,20], which have different plant and soil management practices than the US [24]. Furthermore, many studies that assess the relative growth promotion of native AM fungal in organic cropping systems do not assess the crop response across crop cultivars. Past work on crop varieties or cultivars has found significant variation in varietal response to mycorrhizal inoculation, as well as variation under different abiotic growing conditions [25,26,27]. Additionally, many experiments investigating crop genotypic variation in response to AM fungal inoculation have occurred in greenhouse environments and not in organically managed fields [19,25,28].

To assess how crop cultivars might respond to a native AM fungal amendment under organic growing conditions, five tomato and five pepper varieties were assessed for their response to a common native AM fungal inoculation treatment during a full season of field planting. Crops were grown in conjunction with a local organic farmer using United States Department of Agriculture (USDA) organic growing practices. Crops were transplanted as non-inoculated seedlings, and a native AM fungi consortium was applied at the time of planting. These native fungi were isolated from an old-growth grassland and known to be beneficial to perennial cropping systems [10,26] and native systems [29,30]. Crop productivity was assessed throughout an entire growing season. We hypothesized that native AM fungi would improve crop biomass production for both crop species. Given the strong variation in mycorrhizal responsiveness among crop varietals, we anticipated significant interaction of crop varieties with soil inoculation.

## 2. Materials and Methods

### 2.1. Seedling Germination and Transplanting

Tomato seeds were purchased from Johnny’s (Winslow, ME 04901, USA) of the heirloom tomato varieties Black Krim (lot 62180), Valencia (lot 59702), Brandywine (lot 59111), Green Zebra (lot 38419), and Striped German (lot 61221). Seeds were all heirloom and organic. Seeds were germinated on 15 February 2021 in sterilized peat in a 72-cell tray. Seedlings were watered daily before field transplantation on 12 April 2021.

Bell peppers seed (Islander lot 66044 and Milena lot 60623), jalapeño (Jedi lot 68578 and Mammoth lot 67290), and shishito (Mellow Star lot 66894) were purchased from Speedway (Hall, NY 14463, USA). Seeds were all heirloom and organic. Seeds were germinated in 72-cell trays containing sterilized peat beginning on 15 March 2021. Seedlings were watered daily to field capacity until being transplanted into the field on 12 May 2021.

### 2.2. Experimental Design

This experiment took place on the working organic farm Juniper Hill Farms in Lawrence, Kansas, USA (39.02929, −95.2118) underneath a hoop house, which is an elongated polytunnel made from steel and covered in polyethylene. This hoop house area has been organically managed for more than a decade. Prior to planting, the soil was prepped in the fall via tilling and rolling flat. Soil conditions were 34 ppm P-M2, 32.6 ppm NO^3^, 23 ppm Ca-P, 212 ppm K, 3170 ppm Ca, 225 ppm Mg, 33 ppm Na, 1.6 ppm Zn, 28.9 ppm Fe, 8.3 ppm Mn, 0.5 ppm Boron, and 0.9 ppm Cu. No effort was made to hinder the existing AM fungi present in this field.

Five pepper varieties were each planted along a 100 m row under a single hoop house, with each variety planted the entire row length. Rows were spaced 1.5 m apart and separated by a black weed barrier. One seedling was planted every 0.5 m along the row. Four blocks were established along each row containing 6 pepper plants per treatment. Shishito peppers included three blocks instead of four due to hoophouse constraints. In each block, treatments were arranged by plants inoculated with AM fungi followed by non-inoculated plants so that inoculation treatments were spatially clumped across rows to limit cross-contamination between rows. Treatments within a block and between each of the four blocks were spatially separated by 4 m and were planted with six non-inoculated aisle plants that acted as a buffer and were not included in this experiment. Each week, or as the peppers were ripe, the fruit biomass was harvested from an entire block and the wet biomass was immediately recorded. The entire hoophouse contained 1000 pepper plants.

The tomato varieties were each planted down a single row with the exception being Striped German, which was split across two rows (Section 2.5). Rows were spaced 1.5 m apart and separated by a black weed barrier plastic tarp. One plant was planted every 0.8 m. Blocks were established along each row containing 4 inoculated and 4 non-inoculated tomato seedlings. Black Krim and Pink Brandywine each had four blocks, while the other three tested varieties included three blocks due to constraints in the hoophouse. In each block, treatments were arranged by plants inoculated with AM fungi followed by non-inoculated plants so that inoculation treatments were spatially clumped across rows to limit cross-contamination between rows. Treatments within a block and between each of the four blocks were spatially separated by 4 m that were planted with four non-inoculated aisle plants, which acted as a buffer and were not included in this experiment. Each week, or as tomatoes were ripe, the fruit biomass was harvested, and the wet biomass and tomato number was recorded from each individual plant. After the first harvest, we did not collect additional biomass from two tomato varieties (Valencia and Green Zebra) due to a lack of a consistent fruit set. The entire tomato hoophouse contained 375 tomato plants.

### 2.3. Nutrient and Pest Amendments for Both Tomatoes and Peppers

All the plants were watered daily via a drip irrigation system that ran down each plant row. Each week, all plants were fertilized organically with Proactive Agriculture (119 N Broadway St., Lacygne, KS, USA 66040) products applied via drip irrigation including 5-12-14 Organic (2 lbs. per acre/2.2417 kg per hectare) and 15-1-1 Organic (2 lbs. per acre/2.2417 kg per hectare), High Energy Blend (1 pint per acre/1.23553 L per hectare) MicroPlex Micronutrients (1 pint per acre/1.23553 L per hectare, contains manganese, zinc, iron, boron, sulfur, and humic acid) and Enhanced Coral Calcium (1.5 lbs. per acre/1.68128 kg per hectare). Foliar applications happened 1–2 times per month and included Evergreen (MGK 7325 Aspen Lane North, Minneapolios, MN 55428, 8 oz per acre), Champ WG Agricultural Fungicide (Nufarm Americas Inc. 11901 South Austin Avenue, Alsip, IL, USA 60803 (1.5 lbs. per acre/1.68128 kg per hectare)) and two Proactive Agriculture products (TKO Plus 1 pint per acre/1.23553 L per hectare) and Global Earth TEK (1 pint per acre/1.23553 L per hectare). Each of these products was approved for use in organic agriculture by the United States Department of Agriculture (USDA) or were OMRI listed for use in organic production by meeting USDA National Organic Program standards [31].

### 2.4. Inoculation Treatments for Both Tomatoes and Peppers

The native AM fungi inoculum was created using single-species fungal cultures. The spores for cultures were originally isolated from an old-growth remnant prairie grassland in Lawrence, Kansas (39.04619208°, −95.2050294°) that was located 3.0 km from the farm site. Cultures were grown in 2019 for one year in a sterilized sand:soil mixture (10.15 P ppm via Melich extraction, 7.375 NO^3−^N ppm and 22.2 NO^3−^N ppm via KCl extractions). A native AM fungal community mixture was created by mixing 7 AM fungal species: *Ambispora leptoticha*, *Gigaspora margarita*, *Funneliformis mosseae*, *Rhizophagus clarus*, *Glomus mortonii*, *Rhizophagus diaphanous*, and *Claroideoglomus claroideum*. Past work has shown that these native AM fungal species benefit native prairie plants from this region [32,33,34] and perennial agricultural plant species [10,26]. AM fungal spore density was approximately 13 spores/g or 13,000 spores/kilogram. The inoculum was applied via a “side-dressing approach”. This approach included putting 2 teaspoons (~10 g) of inocula inside an 8 cm deep transplant hole just before placing seedling roots inside that hole and covering both the roots and inocula with nearby soil. Inocula treatments included either living inocula or a sterilized inocula control (non-inoculated) that were killed via autoclaving.

### 2.5. Statistical Analysis

For peppers, the total biomass was log (1 + g) transformed prior to analysis. Proc Mixed was used in the SAS model [35] with average initial size (log initial height (cm) × variety and log initial leaf number × variety), inoculation treatment, pepper variety, and inoculation treatment × pepper variety as fixed effects and block as a random effect. Pepper mass was collected from each block across thirteen weekly or biweekly harvests from 28 June 2021 to 20 October 2021 (Table 1). It should be noted that non-inoculated peppers of the variety Milena in block 3 had to be removed from analysis due to the incorrect pepper variety being planted in this block. Milena pepper did not produce fruit until 28 July 2021. Each pepper/variety block was collected from 6 to 10 times depending on the variety. For analysis of total tomato biomass (log (1 + g)), the total number (log (1 + number)), and the average size of tomato (log (1 + g)), a similar statistical analysis was used as for the peppers, which included the same predictors and block as a random effect. To control for a variety that was planted across two rows, a block × row × inoculation × variety was added as a second random effect. Harvestable tomato biomass and fruit number was collected weekly from 14 August 2021 to 27 September 2021 (Table 1). Because of a lack of tomato production and disease, the tomato varieties Valencia and Green Zebra were not collected until 24 August 2021. At each harvest, the biomass was compiled into a cumulative number prior to being log-transformed. Relative improvement was calculated for the total pepper biomass, total tomato biomass, and average tomato number from the LS means estimates from the SAS model using the following formula:$$100\times \left(\frac{Inoculated\;Plants}{Non-Inoculated\;Plants}\right)-100$$

This reflects the percentage improved (+% values) or hindered (−% values) that AM fungi inoculation provided plants.

## 3. Results

### 3.1. Peppers

Pepper biomass was not significantly affected by the variety or soil treatment in the first three harvests. However, beginning at the fourth harvest, the pepper variety was always significant (*p* < 0.05) for all harvests except one (Table 2). Inoculation treatment was significant or marginally significant at harvest 4 (*p* = 0.057), harvest 5 (*p* = 0.092), harvest 7 (*p* = 0.106), harvest 8 (*p* = 0.060), harvest 12 (*p* = 0.059), and harvest 13 (*p* = 0.077) (Table 2, Figure 1A). Inoculation with native AM fungi consistently improved the total pepper biomass throughout the experiment, resulting in ~+1–8% more biomass depending on the harvest date (Figure 1B). At the final harvest, AM fungal inoculation improved the total pepper mass by 3.7%, which resulted in 3225 g (7.11 lbs.) more peppers on average per block of six plants.

A significant or marginally significant pepper variety X soil inoculation treatment interaction was observed for harvest 4 (*p* = 0.032), harvest 5 (0.038), harvest 7 (0.042), harvest 8 (*p* = 0.022), harvest 9 (*p* = 0.044), harvest 10 (*p* = 0.100), and harvest 12 (*p* = 0.086) (Table 2). Islander bell peppers and Shishito peppers generally benefited from the native AM fungal inoculation throughout the experiment by having greater pepper biomass. At the final harvest, inoculation with native AM fungi resulted in +2782 g (6.13 lbs.) more peppers for Islander bell peppers (Figure 2A) and +10,514 g (23.2 lbs.) more for Shishito peppers (Figure 2B). The Milena bell pepper was strongly mycorrhizally responsive at first, but then it tapered off, with the final biomass being improved +363.8 g (0.80 lbs.) with AM fungal inoculation (Figure 2C). Jedi became more responsive to mycorrhizae at the end of the growing season (harvests 9–12), but biomass was ultimately reduced −343 g (0.76 lbs.) with inoculation (Figure 2D). Mammoth jalapeño peppers did not respond to the mycorrhizae, and the end result was a 1.07% reduction in biomass collected (−310 g/0.80 lbs.) (Appendix A).

### 3.2. Tomatoes

The total cumulative tomato biomass was positively affected by tomato inoculation treatment, and this was especially strong early in the experiment. At the first harvest, inoculation resulted in +253.8 g (0.56 lbs.) more tomato biomass per plant (Table 3, Figure 3A, *p* = 0.031). The average tomato number was twice as high with AM fungi inoculation at the first harvest (Table 3, Figure 3B, *p* = 0.007), with an average of 2.2 tomatoes for inoculated plants and 1.08 tomatoes for non-inoculated plants. There was a marginal effect for larger average tomatoes during the first harvest, where inoculated tomato plants had 35% larger tomatoes (Table 3, Figure 3C, *p* = 0.068) which weighed 48 g more on average. Interactions with the soil × tomato variety were not observed at the first or any harvests (Table 3), and tomato varieties generally all benefited from inoculation (Appendix A).

Across the growing season, native AM fungal inoculation increased the tomato biomass harvested, and this was significant or marginally significant in the first four harvests (Table 3, Figure 4A; harvest 1 *p* = 0.031; harvest 2 *p* = 0.108; harvest 3 *p* = 0.082; harvest 4 *p* = 0.114). This pattern tapered to non-significant trends for the last three harvests 5–7 (*p* = 0.185–0.189). For the last harvests, very few tomatoes were collected, including 37 at harvest 6 and only 10 at harvest 7 in total from among all 134 plants assessed. Whether or not significant at a particular harvest point, tomato biomass was consistently increased due to inoculation throughout all seven harvests that ranged from 44% more tomatoes at harvest 1 to 10% more tomatoes at the final harvest (Figure 4B).

The improved biomass response to inoculation was driven by significant changes in tomato number and not tomato size, which was never significantly different due to inoculation treatments (Table 3). The average tomato number was significantly improved by inoculation with AM fungi beginning at harvest 1 (*p* = 0.007), and soil inoculation remained as either a significant or marginally significant predictor of the number of tomatoes produced throughout the duration of the experiment (Table 3, Figure 4C, *p* ranged from 0.007 to 0.076). The percentage improvement in the average tomato number with inoculation ranged from +69% more tomatoes at harvest 1 to +20% more tomatoes at harvest 7 (Figure 4D). Inoculation resulted in an average of 1.4 more heirloom tomatoes per plant at the end of the experiment. Variety X soil inoculations were not significant for tomato number, meaning that tomato varieties generally responded similarly to inoculation.

## 4. Discussion

Maintaining productive soils in organic agriculture is highly challenging, as fewer options are available for farmers to improve the abiotic and biotic soil conditions of their fields. Arbuscular mycorrhizal (AM) fungi are key members of the soil community that are central to improving soil health and crop productivity. Although AM fungi are commonly present in soils, AM fungal density, diversity, and composition in agricultural environments may be limited by site history [12,13]. Past work assessing native microbial amendments in agriculture has found that inoculation benefits organic cropping systems for corn [16], tropical trees [17], and other crops [18,19,20]. In this trial, five tomato and five pepper varieties were assessed for their response to a common native AM fungal inoculation treatment. Crops were grown in conjunction with a local organic farmer using USDA organic growing methods, which included a hoop house, regular irrigation, and organic fertilizer and pest control applications. Overall, we found support for the hypothesis that adding native mycorrhizal amendments to cropping systems can increase crop yield. Contrary to expectations, we found that native AM fungal inoculations benefited crop varietals similarly.

Our first hypothesis was that native AM fungi would improve crop biomass production for both crop species. Overall, this hypothesis was supported for both peppers and tomatoes. Inoculation with native AM fungi consistently improved total pepper biomass throughout the experiment, and the percentage improvement over the non-inoculated controls was +1–8%, depending on the harvest date. At the final harvest, AM fungal inoculation improved total pepper mass by 3.7%, which resulted in an average of 3225 g (7.11 lbs.) more peppers per block (average of 536 g (1.2 lbs.) per pepper plant). Tomatoes responded to soil inoculation starting at the very first harvest with +69% more tomatoes and +43% more total tomato biomass. Throughout the 13 harvests, inoculated tomatoes consistently produced more tomatoes than their control.

This experiment occurred at a site that used organic fertilizer inputs. Despite the frequent organic nutrient amendment, the addition of biotic amendments of native mycorrhizal fungi improved the pepper and tomato biomass. Additionally, no effort was made at the farm to reduce the presence of resident fungi. These findings support past work indicating that resident fungi in agricultural fields may not perform very efficiently and that amendment with native microbes can boost crop production [18,19,20]. The duration of how long inoculation effects may last in a field will require further investigation. However, given that these fields are planted with annual crops and ploughed each year, which will continuously disturb fungal communities, a yearly application may be required. One past study found that the effects of inoculation persisted for two years [16] for an annual crop. Another study found that AM fungal inoculation can spread meters per year across a field [36]. However, research on both the persistence and spread of inoculation in organic agriculture is lacking.

We anticipated significant interaction of varieties with soil inoculation, where some would prefer mycorrhizal inoculation more than others. This hypothesis was proposed due to the strong variation in mycorrhizal responsiveness among cultivars of certain crop species in past studies [19,25,26,27,28]. This hypothesis was not supported for tomatoes and only partially supported for peppers. Tomato varieties seemed to generally respond positively to mycorrhizal inoculation, as there was never a significant variety X soil inoculation interaction for any metric in any of the harvest periods. We observed significant variety X soil inoculation interactions for peppers at 4 of the 13 harvests, particularly in the middle of the experiment for harvests 4–9. However, this pattern was not observed in the beginning or during the last four pepper harvests. This pattern appeared to be driven by the jalapeño varieties tested being less responvie to inoculation. Mammoth jalapeño never responded to mycorrhizal fungi, and Jedi jalapeño only responded positively to AM fungi towards the end of the trial. In contrast, Shishito and Islander bell peppers consistently benefited more from native AM fungal inoculation throughout the experiment by producing more peppers.

It should be noted that our native inocula was sourced from a nearby old-growth native grassland system and not sourced from an agricultural field. Thus, the fungi we selected may be functionally different from those in agricultural lands. Past work has found that organic agricultural practices can improve AM fungal community diversity nearly twice as much as conventional agriculture [3,37]. However, the dominant species in organic agricultural systems may not overlap with those in conventional agricultural systems [3,37] or with native systems [38]. Furthermore, fungi that persist in organic agriculture may not actually be beneficial to cropped plants [39], potentially because agricultural fungi have already been selected to have more ruderal and non-beneficial traits, including heavy investment in fungal reproduction at the expense of nutrient acquisition processes that can beneficial crop hosts [40]. Thus, some may argue that there may be little evidence that farmers should consider practices to conserve fungi when managing crops [41]; this may be because the diversity and functionality of fungi in agricultural soils may already be too damaged to repair using soil conservation practices. Here, we show that adding a consortium of native fungi known to be beneficial can have season-long benefits in organic agricultural systems. Past work has shown that applications of native AM fungi sourced from old-growth habitats can also benefit perennial cropping systems [10,26] and native system restorations [29,30]. Taken together, these data reveal a pattern where applications of native fungi sourced from old-growth ecosystems can benefit plants in a multitude of environments, from restoration to organic agriculture, due to the increased beneficial function of the fungi isolated from old-growth native habitats. Future work should assess how widespread this pattern is across the globe.

## 5. Conclusions

This study highlights the potential benefits of using native AM fungi as a soil amendment in organic farming systems to improve pepper and tomato productivity under organically managed soils. Tomatoes benefited from inoculation beginning at the first harvest, with almost twice the number of harvested tomatoes per plant, and most pepper varieties also benefited from inoculation in the early harvests. Thus, inoculation can provide organic farmers with a boost to crop production early in the season, potentially giving them a competitive edge in the spring and early summer market. Because inoculation generally improved fruit production for both crops, this study highlights that native mycorrhizal inoculations can be used in organic soils that are disturbed annually. This work contributes to a growing body of literature suggesting that disturbed soils harbor ineffective AM fungal symbionts, but that plant–fungal relationships can be repaired with the addition of beneficial fungal amendments. From an application perspective, the “side-dressing” approach to inoculation incorporated by this farmer was easy to do at planting and could be a viable way to introduce mycorrhizal fungi at farms of many scales and sizes.

## Figures and Tables

**Figure 1 microorganisms-11-02012-f001:**
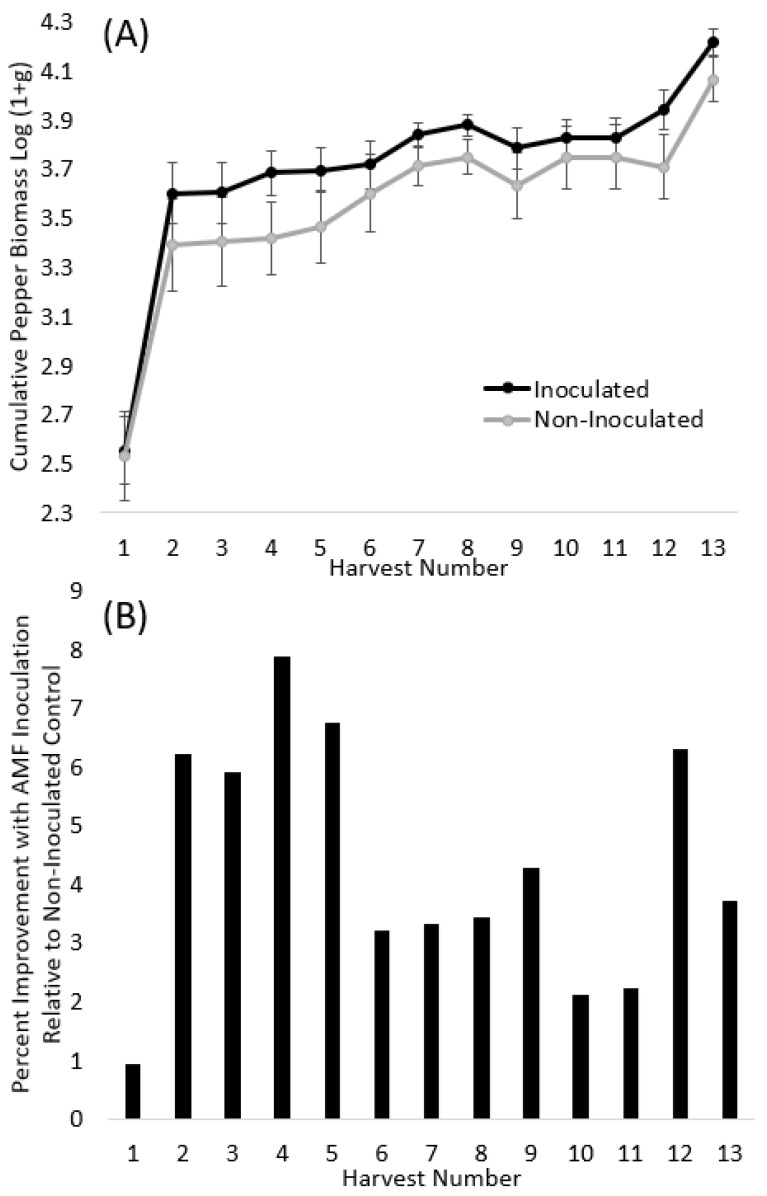
(**A**) Log (1 + g) transformed cumulated pepper biomass across all thirteen harvests for the inoculated (black line) and non-inoculated (grey line) plants. The points are LS means from the proc mixed model, and the error bars are standard error. (**B**) Relative improvement with native AM fungi inoculation across each of the thirteen harvests ranged from +1 to 8%.

**Figure 2 microorganisms-11-02012-f002:**
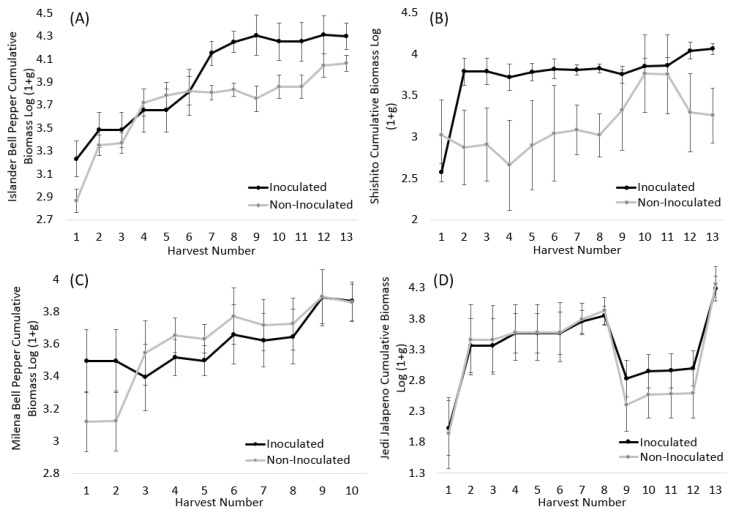
Cumulative log (1 + g) transformed pepper biomass for (**A**) Islander Bell, (**B**) Shishito, (**C**) Milena bell pepper, and (**D**) Jedi jalapeño across all thirteen harvests for inoculated (black line) and non-inoculated (grey line) plants. The points are LS means, and the error bars are standard error outputs from the proc mixed model.

**Figure 3 microorganisms-11-02012-f003:**
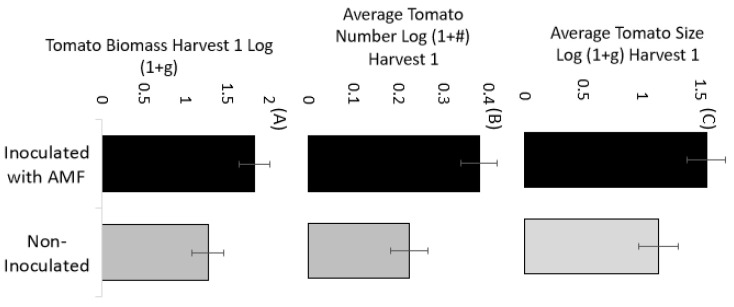
(**A**) Total biomass, (**B**) tomato number, and (**C**) average tomato size at the end of the first harvest with (black bars) and without (grey bars) inoculation. Inoculation greatly improved tomato biomass and tomato number beginning at this first harvest. The bars are LS means from the proc mixed model and the error bars are standard error.

**Figure 4 microorganisms-11-02012-f004:**
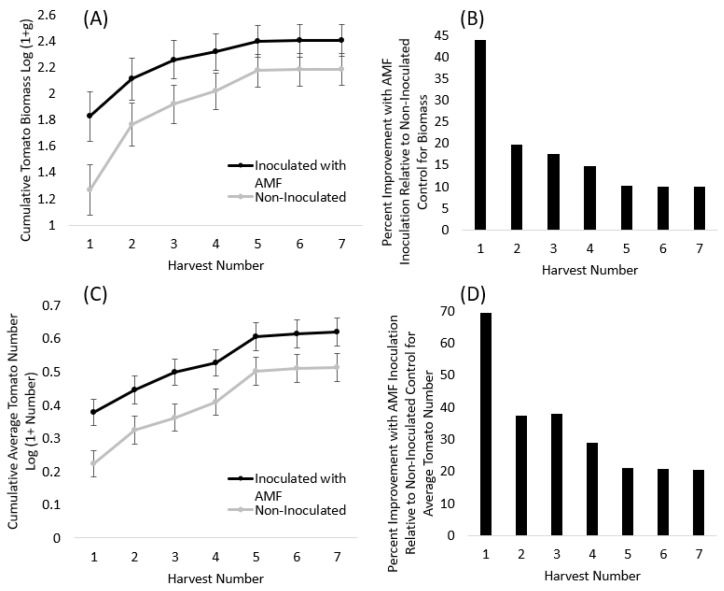
(**A**) Tomato fruit biomass (log (1 + g)) and the (**B**) relative improvement in tomato biomass due to native AM fungal inoculation over the seven harvests. (**C**) Tomato fruit number (log (1 + number)) and the (**D**) relative percent improvement in the number of tomatoes due to inoculation across all seven harvests. The points are LS means from the proc mixed model and the error bars are standard error. Lines represent inoculated (black line) and non-inoculated (grey line) plants. Bars represent relative improvement (% increase) calculated from the LS means of the proc mixed model.

**Table 1 microorganisms-11-02012-t001:** Harvest number and date for tomatoes and peppers.

Harvest Number	Pepper Harvest Date	Tomato Harvest Date
1	28 June 2021	14 August 2021
2	15 July 2021	24 August 2021
3	27 July 2021	31 August 2021
4	28 July 2021	7 September 2021
5	15 August 2021	14 September 2021
6	21 August 2021	21 September 2021
7	26 August 2021	27 September 2021
8	10 September 2021	
9	25 September 2021	
10	28 September 2021	
11	7 October 2021	
12	12 October 2021	
13	20 October 2021	

**Table 2 microorganisms-11-02012-t002:** Average cumulative pepper biomass. F-value (F) and *p*-value (*p*) outputs from the proc mixed model. Harvests 1–13 represent the harvest dates in Table 1. Harvest 1–3 have lower numerator degrees of freedom because the Milena pepper did not fruit until harvest 4 and was excluded until harvest 4.

Predictors	N	D	Harvest 1		Harvest 2				Harvest 3			
			**F**	* **p** *	**F**		* **p** *		**F**		* **p** *	
Initial Height × Variety	4	11	1.72	0.216	0.8		0.55		0.76		0.573	
Initial Leaf × Variety	4	11	0.57	0.69	1.32		0.323		1.27		0.339	
Soil Treatment	1	11	0.03	0.875	1.82		0.204		1.71		0.218	
Variety	3	11	0.99	0.432	1.36		0.306		1.33		0.314	
Soil Treatment × Variety	3	11	1.96	0.179	1.78		0.208		1.66		0.234	
**Predictors**	**N**	**D**	**Harvest 4**		**Harvest 5**		**Harvest 6**		**Harvest 7**		**Harvest 8**	
			**F**	* **p** *	F	* **p** *	F	* **p** *	F	* **p** *	F	* **p** *
Initial Height × Variety	5	14	0.96	0.474	0.75	0.599	0.52	0.76	1.78	0.183	3.22	0.038
Initial Leaf × Variety	5	14	1.97	0.146	1.78	0.182	0.66	0.657	3.01	0.047	5.41	0.006
Soil Treatment	1	14	4.3	0.057	3.27	0.092	0.7	0.418	2.99	0.106	4.19	0.06
Variety	4	14	3.33	0.041	3.06	0.052	0.69	0.608	3.64	0.031	5.97	0.005
Soil Treatment × Variety	4	14	3.61	0.032	3.43	0.038	0.72	0.591	3.31	0.042	4.03	0.022
**Predictors**	**N**	**D**	**Harvest 9**		**Harvest 10**		**Harvest 11**		**Harvest 12**		**Harvest 13**	
			**F**	* **p** *	**F**	* **p** *	F	* **p** *	F	* **p** *	F	* **p** *
Initial Height × Variety	5	14	2.36	0.094	2.02	0.139	2.07	0.131	2.43	0.087	2.75	0.062
Initial Leaf × Variety	5	14	4.16	0.016	3.59	0.027	3.34	0.034	3.7	0.024	3.21	0.039
Soil Treatment	1	14	1.82	0.199	0.49	0.493	0.53	0.478	4.25	0.058	3.65	0.077
Variety	4	14	4.67	0.013	4.16	0.02	3.93	0.024	4.45	0.016	6.61	0.003
Soil Treatment × Variety	4	14	3.25	0.044	2.4	0.1	2.17	0.126	2.55	0.086	1.75	0.195

**Table 3 microorganisms-11-02012-t003:** Average cumulative tomato (A) biomass, (B) number, and (C) average tomato size. F-value (F) and *p*-value (*p*) outputs from the Proc Mixed model. Harvests 1–7 represent the harvest dates in Table 1.

Predictors			Harvest 1		Harvest 2		Harvest 3		Harvest 4		Harvest 5		Harvest 6		Harvest 7	
**(A) Total Tomato Biomass**	**N**	**D**	**F**	* **p** *	**F**	* **p** *	**F**	* **p** *	**F**	* **p** *	**F**	* **p** *	**F**	* **p** *	**F**	* **p** *
Initial Height × Variety	5	86	1.38	0.239	2.53	0.035	0.88	0.496	0.42	0.836	0.41	0.840	0.4	0.845	0.4	0.849
Initial Leaf × Variety	5	86	1.4	0.231	1.51	0.196	2	0.087	2.21	0.061	2.19	0.063	2.2	0.062	2.19	0.062
Variety	4	19	2.17	0.111	3.04	0.043	2.33	0.093	0.93	0.465	0.58	0.681	0.58	0.682	0.57	0.687
Soil Treatment	1	19	5.44	0.031	2.84	0.108	3.37	0.082	2.75	0.114	1.9	0.185	1.86	0.188	1.86	0.189
Soil Treatment × Variety	4	19	0.92	0.472	0.32	0.858	0.13	0.971	0.08	0.988	0.23	0.917	0.23	0.917	0.23	0.918
**(B) Total Tomato Number**	**N**	**D**	**F**	* **p** *	**F**	* **p** *	**F**	* **p** *	**F**	* **p** *	**F**	* **p** *	**F**	* **p** *	**F**	* **p** *
Initial Height × Variety	5	86	1.17	0.328	1.69	0.145	0.97	0.438	0.98	0.435	0.56	0.733	0.51	0.771	0.5	0.779
Initial Leaf × Variety	5	86	1.18	0.327	1.39	0.236	1.73	0.135	2.5	0.036	1.24	0.299	1.41	0.229	1.4	0.233
Variety	4	19	1.82	0.167	2.66	0.064	2.27	0.099	2.41	0.085	1.08	0.394	1.18	0.352	1.16	0.360
Soil Treatment	1	19	9.25	0.007	5.28	0.033	7.36	0.014	5.29	0.033	3.72	0.069	3.54	0.075	3.52	0.076
Soil Treatment × Variety	4	19	1.27	0.318	0.77	0.558	0.22	0.922	0.29	0.880	0.49	0.741	0.48	0.753	0.46	0.766
**(C) Average Tomato Size**	**N**	**D**	**F**	* **p** *	F	* **p** *	F	* **p** *	F	* **p** *	F	* **p** *	F	* **p** *	F	* **p** *
Initial Height × Variety	5	86	1.31	0.266	2.43	0.042	0.64	0.673	0.38	0.864	0.71	0.616	0.68	0.641	0.68	0.639
Initial Leaf × Variety	5	86	1.37	0.245	1.42	0.226	1.92	0.099	1.89	0.104	2.17	0.064	2.14	0.068	2.14	0.068
Variety	4	19	2.06	0.127	2.64	0.066	1.79	0.172	0.36	0.836	0.35	0.844	0.31	0.865	0.32	0.862
Soil Treatment	1	19	3.73	0.068	1.58	0.224	1.32	0.265	1.28	0.273	0.71	0.411	0.69	0.417	0.69	0.418
Soil Treatment × Variety	4	19	1.07	0.399	0.59	0.673	0.31	0.865	0.28	0.886	0.48	0.747	0.47	0.754	0.48	0.751

## Data Availability

The data presented in this study are available in Appendix A.

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
