# Peer review of "Crop Productivity Boosters: Native Mycorrhizal Fungi from an Old-Growth Grassland Benefits Tomato (Solanum lycopersicum) and Pepper (Capsicum annuum) Varieties in Organically Farmed Soils"

_microorganisms, 2023, doi:10.3390/microorganisms11082012_

Round 1
Reviewer 1 Report
The manuscript entitled “Crop Productivity Boosters: Native Mycorrhizal Fungi from an Old-Growth Grassland Benefits Tomato (Solanum lycopersicum) and Pepper (Capsicum annuum) Varieties in Organically Farmed Soils” presents a very important study
The manuscript is very extensive and written comprehensively.
In my opinion, small corrections should be made
I don't know if we can talk about agriculture or organic farming here.
There is no indication whether the seed material of peppers and tomatoes was exclusively organic, it is not indicated.
In organic farming: only organic seed
The principle of an organic farm is:
lack of any chemicals in plant protection and mechanical weed control;
fungicide has been applied! Champ WG Agricultural Fungicide (line 147)
What is organic watering? What is Proactive Agriculture? The reader cannot guess, he must have it described.
MicroPlex Micronutrients does not describe what micronutrients it contains
Author Response
Reviewer #1
The manuscript entitled “Crop Productivity Boosters: Native Mycorrhizal Fungi from an Old-Growth Grassland Benefits Tomato (Solanum lycopersicum) and Pepper (Capsicum annuum) Varieties in Organically Farmed Soils” presents a very important study. The manuscript is very extensive and written comprehensively. In my opinion, small corrections should be made
I don't know if we can talk about agriculture or organic farming here.
There is no indication whether the seed material of peppers and tomatoes was exclusively organic, it is not indicated.
In organic farming: only organic seed
Yes, we used organic seed. We have revised the manuscript to include that “seeds were all heirloom and organic. Additionally, we add the below details given your comments below.
The principle of an organic farm is:
- lack of any chemicals in plant protection and mechanical weed control;
- fungicide has been applied! Champ WG Agricultural Fungicide (line 147)
- What is organic watering? What is Proactive Agriculture? The reader cannot guess, he must have it described.
- MicroPlex Micronutrients does not describe what micronutrients it contains
We thank the reviewer for their comments. Considering #1, we understand that “organic agriculture” may differ by region. We state that the site followed regionally appropriate and approved organic methods. These experiments took place within a working organically certified farm in Lawrence, Kansas, USA. Considering #2, the United States of Agriculture does allow for use of chemicals and even fungicide in organic practices. We now state in the methods that all products were approved by the USDA or listed as OMRI organically certified and give a reference for OMRI. With regards to #3, Proactive Agricultural is an agricultural company. We forgot to capitalize the “A.” We revise the manuscript to read “Proactive Agriculture (119 N Broadway St, Lacygne, KS, USA 66040). We find no references to organic watering in the manuscript. We state, “All plants were watered daily via a drip irrigation system that ran down each plant row.” We revise the manuscript to read “MicroPlex Micronutrients contains manganese, zinc, iron, boron, sulfur and humic acid.”
Reviewer 2 Report
Dear authors, the manuscript is clear written and present the results obtained through the conducted research.
As a suggestion, try to organize the tables presented, especially Table 1 and 2. They do not fit into the manuscript.
Author Response
Reviewer #2
Dear authors, the manuscript is clear written and present the results obtained through the conducted research.
As a suggestion, try to organize the tables presented, especially Table 1 and 2. They do not fit into the manuscript.
We thank reviewer #2 for the suggestion. We will leave the manuscript as is and will trust the copy editor will make suggestions on this for us to follow up during article processing.